# Contribution of the Degeneration of the Neuro-Axonal Unit to the Pathogenesis of Multiple Sclerosis

**DOI:** 10.3390/brainsci7060069

**Published:** 2017-06-18

**Authors:** Hannah E. Salapa, Sangmin Lee, Yoojin Shin, Michael C. Levin

**Affiliations:** 1Department of Anatomy and Cell Biology, CMSNRC (Cameco MS Neuroscience Research Center), University of Saskatchewan, Saskatoon, SK S7N0Z1, Canada; hes763@mail.usask.ca; 2Veterans Administration Medical Center, Memphis, TN 38104, USA; salee@uthsc.edu (S.L.); yshin2@uthsc.edu (Y.S.); 3Department of Neurology, University of Tennessee Health Science Center, Memphis, TN 38104, USA; 4Department of Neurology, University of Saskatchewan, Saskatoon, SK S7N0Z1, Canada

**Keywords:** multiple sclerosis, RNA binding protein, neurodegeneration, axonal damage, hnRNP A1

## Abstract

Multiple sclerosis (MS) is a demyelinating, autoimmune disease of the central nervous system. In recent years, it has become more evident that neurodegeneration, including neuronal damage and axonal injury, underlies permanent disability in MS. This manuscript reviews some of the mechanisms that could be responsible for neurodegeneration and axonal damage in MS and highlights the potential role that dysfunctional heterogeneous nuclear ribonucleoprotein A1 (hnRNP A1) and antibodies to hnRNP A1 may play in MS pathogenesis.

## 1. Multiple Sclerosis (MS)

Multiple sclerosis (MS) is a demyelinating, autoimmune disease of the central nervous system (CNS). Over two million affected individuals world-wide, typically diagnosed in young adulthood, makes MS the most debilitating neurological disease in this population. Symptoms associated with MS, such as fatigue, impaired coordination, and spasticity, limit the ability of people to function properly, which endows a financial burden on both the patient and their family. The majority of patients are initially diagnosed with relapsing remitting MS (RRMS) where symptoms develop and are followed by a period of recovery or remission with no symptoms [1,2,3]. A total of 50% of RRMS patients gradually advance to having secondary progressive MS (SPMS) where symptoms steadily increase and worsen with rare periods of recovery [1,2,3,4]. Approximately 5% of people develop primary progressive MS (PPMS) where patients experience gradual disease worsening with no periods of recovery [1,3]. Additional clinical subtypes include radiologically isolated syndrome (RIS) where lesions are discovered on MRI without the presentation of any symptoms and clinically isolated syndrome (CIS) where a patient experiences an initial clinical episode consistent with MS concurrent with an MRI suggestive of MS [4]. Despite the differences in onset, relapse rate, and initial subtype diagnosis, many patients with MS will progress to a stage of irreversible disability [5].

One of the most recognizable pathological features of MS is the plaque or lesion that is evident in vivo on MRI brain scans. These plaques, depending on their stage, can contain activated lymphocytes, microglia, and myelin debris from macrophage degradation [6]. Oligodendrocytes often attempt to remyelinate damaged axons, which leads to shadow plaques containing partially remyelinated axons [1]. Inflammation, plaques, and disease progression vary between individuals and may not necessarily correlate with disease severity. For example, white matter atrophy rates have been shown to be similar across each subtype of disease; however, there are vast differences in grey matter atrophy across disease subtypes [7]. Significantly more grey matter atrophy is observed in SPMS patients and those with higher expanded disability status scale (EDSS) scores as opposed to RRMS patients or those with lower EDSS scores [8]. These findings and other incongruences between disease state and pathology suggest that inflammation and neurodegeneration, including axonal damage, likely transpire simultaneously but independently during disease [9,10]. Axonal and neuronal injury exists in the absence of demyelination, with no correlation between plaque location and axonal loss in spinal cord long tracts (e.g. corticospinal tract and posterior columns) [11]. Additionally, focal axonal degeneration starts with focal swellings, which are observed in myelinated axons [12]. These differences suggest that neurodegeneration is a better correlate of disease severity than demyelination. Clearly, neurodegeneration in MS takes place throughout disease [13,14,15] and although neurodegeneration is known to cause permanent disability in MS, research regarding the underlying mechanisms is in its infancy.

The degeneration of axons is generally classified as either Wallerian degeneration or “dying back” degeneration. Wallerian degeneration occurs when the axon located distally to a site of injury begins to degenerate in an anterograde fashion. These injuries result in a disconnect between the neuronal cell body and distal portion of the axon. Dying back, on the other hand, is axonal degeneration from the distal end in a retrograde manner with degeneration of the axon happening before cell body loss. Both Wallerian and “dying back” degeneration are characterized by axonal dystrophy, which is visualized as axonal spheroids and varicosities, which have been found in MS lesions [16]. Axonal degeneration observed in MS has been most commonly categorized as Wallerian degeneration [17]. “Dying back” axonopathy has been observed in hereditary spastic paraplegia (HSP), a disease clinically similar to MS [18,19,20].

A number of mechanisms underlying neurodegeneration in neurologic diseases have been proposed. For instance, in amyotrophic lateral sclerosis (ALS), axonal transport deficits and mitochondrial dysfunction have been observed in ALS-mutant mice [21,22]. Additional evidence suggests that the interplay between mitochondria and impaired axonal transport leads to degeneration in Alzheimer’s disease [23]. Dysfunctional mitochondria have also been shown to play a role in the pathogenesis of Parkinson’s Disease [24]. More recent findings suggest a role for RNA binding proteins in neurologic disease [25,26,27,28,29]. RNA binding proteins are responsible for regulating RNA homeostasis, also known as “ribostasis”. The mislocalization of these proteins from nucleus to cytoplasm and the formation of stress granules are key pathological features of RNA binding proteins in ALS, frontotemporal dementia (FTD), and spinal muscular atrophy (SMA) [25,26,27,28,30,31]. In MS, disrupted sodium and calcium ion channel dynamics [32,33], axonal transport deficiencies [9,34,35,36], mitochondrial dysfunction [37,38,39,40], and oxidative stress [40,41] contribute to neuronal and axonal damage. As in other diseases, these mechanisms may work in tandem to contribute to neurodegeneration and axonal damage as opposed to independently (Figure 1).

Axonal transport is an essential function for maintenance of neuronal health, and has long been implicated in neurodegenerative conditions. The neuronal cytoskeleton is composed of microtubules, neurofilaments, and actin filaments. Microtubules provide a track system for the movement of cargo along the axons by the motor proteins kinesin and dynein. Kinesins mediate anterograde transport, moving organelles or vesicles from the soma to the synapse or membrane while dynein molecules are involved in retrograde axonal transport to move cargo toward the cell body (Figure 1). Evidence of disrupted axonal transport and axonal damage is observed in MS post-mortem tissue through staining with amyloid precursor protein (APP), a protein involved in “fast” anterograde transport due to its ability to mediate interactions between cargo and kinesin proteins [42]. APP staining can be seen in acute MS brain lesions [34], axonal swellings in demyelinated plaques [35], and in normal appearing white matter of acute MS cases [9]. Furthermore, impaired transport of organelles is a prominent early feature in inflammatory MS-like lesions [16]. There is further support for this concept in experimental autoimmune encephalomyelitis (EAE) mice where compromised mitochondrial transport is an early event in EAE lesions [35]. Disruption of axonal transport, both retrograde and anterograde, has also been observed prior to demyelination in EAE mice [36]. Researchers used genetically engineered mice expressing fluorescent markers in mitochondria and peroxisomes under the *Thy1* promoter. After inducing EAE, researchers employed two-photon imaging to measure retrograde and anterograde transport of both organelles along axons and found decreased transport in swollen axons as well as in normal appearing axons around lesion areas [36]. These findings in the EAE model as well as those with APP staining in the MS cortex are evidence that loss of axonal transport could be an early first step towards axonal degeneration prior to demyelination.

An essential component for axonal transport is energy, in the form of ATP, which is produced by mitochondria. There is an increasing body of work supporting a role for mitochondrial dysfunction in neurodegenerative diseases [50]. Mitochondria are responsible for maintaining a cell’s energy production in the form of ATP generated by the respiratory chain complex. In addition to producing ATP, mitochondria also function to produce amino acids, maintain calcium homeostasis, and the modulation of reactive oxygen species (ROS). It is therefore understandable that perturbation of mitochondrial processes could result in neuronal dysfunction, decreased viability, and even apoptosis leading to neuronal loss and degeneration. Loss of neurons in the MS patient cortex is commonly observed [51,52]. Incidentally, a decrease in mitochondrial electron transport gene expression is also observed in MS brain tissues, suggesting that mitochondrial dysfunction could be contributing neuronal loss in patients [38]. Furthermore, mitochondrial DNA (mtDNA) accumulates deletions in the grey matter of SPMS patients irrespective of lesions [12,39]. These damaged mitochondria could influence anterograde transport leading to axonal transport deficits [37] and compromised mitochondrial transport is an early change in inflammatory EAE lesions [35]. In addition to compromising axonal transport, dysmorphic swollen mitochondria lead to increased ROS and reactive nitrogen species (RNS) concentrations and therefore, a release of proapoptotic mediators [12].

In addition to these distinctive hypotheses regarding neurodegeneration, there is also evidence for the contribution of antibodies to neurodegeneration in MS. The inflammatory environment in MS primarily consists of an initial T cell infiltrate along with activated macrophages and microglia with another T- and B-cell infiltrate after myelin has been broken apart [13]. The invading T-cells consist of both CD4+ and CD8+ T-cells, however, data suggests that CD8+ may have a more profound effect especially during the later phases of disease [53,54]. Natural killer (NK) cells may also be present in the inflammatory milieu and have been shown to play both protective and deleterious roles in MS [55]. Lymphocytes are present during active demyelination, however, antibody producing plasma cells are more evident in SPMS and PPMS patients, suggesting a role for autoantibodies in disease progression [14]. Autoantibodies to myelin antigens such as myelin oligodendrocyte glycoprotein (MOG), myelin basic protein (MBP), and myelin proteolipid protein (PLP) have been identified. MOG antibodies have been shown to play a primary role in the demyelination of axons as opposed to degeneration [43] through complement cascade activation [44,45] as two myelin proteins bind C1 directly to activate the complement cascade [45]. For example, C3d immunoreactivity is seen in areas of partly demyelinated axons as well as in active lesions [56]. On the other hand, antibodies to non-myelin antigens such as neurofascin, neurofilament, and KIR 4.1 (a glial potassium channel) have been shown to contribute to axonal and neuronal injury [46,47,48,49]. Furthermore, the injection of neurofascin antibodies into EAE mice also leads to axonal injury [48]. Although antibodies to myelin and non-myelin antigens seem to have different effects on myelination and axonal damage, respectively, they may have a common mechanism of action through activation of the complement system [46,47,48] and could both be responsible for the pathology observed in MS tissue. Furthermore, progressive patients have IgG-containing plasma cells in their meninges and throughout the brain, which remain even after T- and B-cell levels decrease [14,57]. Clearly, the humoral response is an important contributing factor to axonal damage but may be particularly influential in patients with progressive disease.

## 2. hnRNP A1 and RNA Metabolism

Support for a role of antibodies to non-myelin, neuronal antigens in MS pathogenesis, specifically neurodegeneration, is strong [14,46,48,49,57,58]. Data from our lab also provides compelling evidence to strengthen this hypothesis. Initial experiments from our lab showed that IgG from human T-lymphotrophic virus type 1 associated myelopathy/tropical spastic paraparesis (HAM/TSP) patients, a disease clinically similar to progressive MS, immunoreacted with a 33 kDa protein from isolated human brain neurons on a Western blot [59]. This protein was identified as heterogeneous nuclear ribonucleoprotein A1 (hnRNP A1) [59]. hnRNP A1 is an RNA binding protein that performs a multitude of functions related to ribostasis, including mRNA transport, pre-mRNA processing, and translation [60]. IgG from HAM/TSP patients was shown to preferentially react with areas commonly damaged in HAM/TSP, such as neurons and axons throughout the corticospinal system [61]. The immunodominant epitope of hnRNP A1 recognized by HAM/TSP IgG was identified as an amino acid sequence (AA 293-GQYFAKPRNQGG-304) within the M9 nuclear localization sequence [62]. The “M9” area is required for the nucleocytoplasmic transport of hnRNP A1 [63]. HAM/TSP and progressive forms of MS show similarities and as such, it was hypothesized that MS patient IgG would also react with hnRNP A1, indicating the development of antibodies against this RNA binding protein (RBP). MS patients were found to make antibodies to hnRNP A1, specifically to the same M9 epitope as HAM/TSP patients [64,65]. Healthy controls and patients with Alzheimer’s disease were examined as controls and were found to show no immunoreactivity to hnRNP A1 [64].

Because antibodies to other non-myelin antigens, such as neurofascin, have been shown to worsen EAE and lead to axonal damage, we hypothesized that anti-hnRNP A1-M9 antibodies, which recognize the same immunodominant epitope as MS patient IgG, might show similar effects. Neurons were exposed to control antibodies as well as hnRNP A1-M9 antibodies. Anti-hnRNP A1-M9 exposure led to neurodegeneration and neuronal death [64]. Microarray analyses comparing anti-hnRNP A1-M9 antibodies to both control IgG and untouched neuronal cells revealed altered RNA expression in the anti-hnRNP A1-M9 antibody condition [64]. Interestingly, some of the genes affected by the anti-hnRNP A1-M9 antibodies included the spinal paraplegia genes (SPGs) implicated in the pathogenesis of hereditary spastic paraplegia (HSP), which clinically mimics HAM/TSP and progressive MS. Specifically, it identified spastin (SPG4), paraplegin (SPG7), and spartin (SPG20) [64]. Furthermore, anti-hnRNP A1-M9 antibodies also altered expression of axonal transport RNAs, including kinesin family member 5 (KIF5C) as well as a number of genes related to hnRNP A1’s function in “ribostasis” [64].

We sought to determine whether anti-hnRNP A1-M9 antibodies had an effect on hnRNP A1’s ability to bind its target RNA (which is bound via the RNA binding domains). Using the RNA Binding Protein Data Base (RBPDB.com), we determined that spastin (SPG4) contains a 100% binding sequence (NM_014946, b.3282-3288) match with the hnRNP A1 binding sequence while SPG7 and SPG20 showed lesser degrees of RNA sequence alignment. By using RNA immunoprecipitation, we found that SP4 and SPG7 bound hnRNP A1 while SPG20 did not [3,66]. Furthermore, SPG4 and SPG7 levels were remarkably decreased in neuronal cells that had been exposed to anti-hnRNP A1-M9 antibodies but not in cells exposed to control antibodies [66]. Because anti-hnRNP A1-M9 antibodies had led to neuronal death, loss of neuronal processes, apoptosis, and mislocazation of hnRNP A1 from the nucleus to the cytoplasm [64,66,67], we hypothesized that the immunodominant anti-hnRNP A1-M9 antibodies might impact the EAE disease course in mice.

To test this hypothesis, we induced EAE in mice and upon the first sign of disease, we injected anti-hnRNP A1-M9 antibodies, control antibodies, and phosphate buffered saline (PBS) three times for a total of 300 micrograms of antibody. We clinically scored animals and approximately 11 days following the first injection, animals injected with anti-hnRNP A1-M9 antibodies showed significantly higher clinical scores, indicating worsened disease [68]. Subsequent staining with Fluoro Jade C showed preferential neurodegeneration in the anti-hnRNP A1-M9 animals in the deep white matter of the cerebellum and the distal ventral spinocerebellar tract (VSCT) as it enters the cerebellum [68]. The cell bodies of the VSCT, an afferent pathway, lie in laminae VII, VIII, and IX of the lumbosacral spinal cord. Recent studies show entry of T-cells happens in this region early in EAE [69,70]. This pattern of neurodegeneration, in which axonal injury follows a distal to proximal pattern, suggests a “dying back” axonopathy, which is commonly observed in HSP [18,19]. Furthermore, these animals developed spasticity (a major clinical feature in MS patients) in their hind limbs whereas those injected with control antibodies or PBS did not.

The development of spasticity, the interaction between hnRNP A1 and SPG4 and SPG7, and the “dying back” axonopathy suggest a similar or shared mechanism of pathology between HSP and progressive MS. In HSP, mutations within SPG4 account for the majority of cases. Spastin is a member of the AAA protein family with multiple isoforms (M1, M87) that has microtubule severing functions. The severing of microtubules by spastin is crucial for efficient microtubule transport and mutations lead to loss of microtubule-severing activity and distal axonal end degeneration [18]. The M1 spastin isoform (68 kDa) is only detectable in the adult mouse spinal cord whereas the M87 (60 kDa) isoform is more widely distributed and more abundant [18]. The presence of the M1 isoform strongly correlates with axonal degeneration in HSP, suggesting a gain of function mechanism due to perturbed alternative splicing mechanisms [71]. Mutations in either isoform alter microtubule dynamics and lead to the formation of toxic aggregates [18]. Mutations within SPG7 also account for a smaller portion of HSP cases. SPG7 plays a role in the inner mitochondrial membrane and cultured myoblasts from patients with SPG7 mutations show defects in respiratory chain function [72].

Because mutations in RNA binding proteins have been shown to lead to other neurological diseases [26,29], we wanted to determine whether MS patients had mutations in hnRNP A1. DNA was isolated from peripheral blood monocytes (PMBC) from each subtype of MS patient and PCR was performed to isolate human hnRNP A1 genomic DNA containing exons 8 and 9. Following amplification, genomic DNA was cloned into vectors and sequenced. PPMS patients had a greater number of novel somatic nucleotide variants, which when translated into protein resulted in more amino acid substitutions than RRMS, SPMS, or healthy controls [73]. These mutations were cloned into expression vectors and transfected into SKNSH cells and stained for hnRNP A1 as well as stress granules. Mutant forms of hnRNP A1, as opposed to wildtype, showed mislocalization from the nucleus to the cytoplasm as well as the formation of stress granules [73], which is similar to pathogenic features observed in other neurologic diseases involving RNA binding proteins [24,27]. Furthermore, exposing SKNSH cells to anti-hnRNP A1-M9 antibodies also leads to mislocalization and the formation of stress granules [66,67]. This suggests that hnRNP A1 dysfunction, either related to an autoimmune or genetic mechanism, may contribute to MS pathogenesis in a manner similar to other diseases.

Taken together, these studies suggest that hnRNP A1, an RNA binding protein, may show pathogenic features, such as mislocalization and stress granule formation, in MS that are similar to other neurologic diseases. After exposure to anti-hnRNP A1 antibodies, hnRNP A1 mislocalizes to the cytoplasm of cells. This mislocalization is also observed in [74]. The effect of anti-hnRNP A1-M9 antibodies on neuronal cell lines and RNA targets in vitro suggests that antibodies may be altering endogenous hnRNP A1 functions by interrupting normal ribostasis such as mRNA binding. If hnRNP A1 does not properly bind RNA targets, such as SPG4 or KIF5C in vivo, this could disrupt normal functioning of these proteins resulting in impaired axonal transport or the development of spasticity. Additionally, anti-hnRNP A1-M9 antibodies, in the setting of a pro-inflammatory environment (EAE), lead to the development of hind limb spasticity and neurodegeneration, including axonal dying back, phenotypes both observed in HSP and MS [20,75,76]. Further understanding the mechanisms and consequences of dysfunctional hnRNP A1 and anti-hnRNP A1 antibodies could lead to the development of better therapies that alleviate symptoms, such as spasticity.

## 3. Conclusions

Several factors contribute to axonal damage and neurodegeneration in MS. Impaired fast axonal transport, the release of reactive oxygen species, mitochondria dysfunction, the redistribution of RNA binding proteins from their normal nuclear location, and antibodies to non-myelin antigens along with pro-inflammatory events occur during MS. A combination of these events, as opposed to one in particular, may be responsible for neuronal and axonal damage observed in disease.

## Figures and Tables

**Figure 1 brainsci-07-00069-f001:**
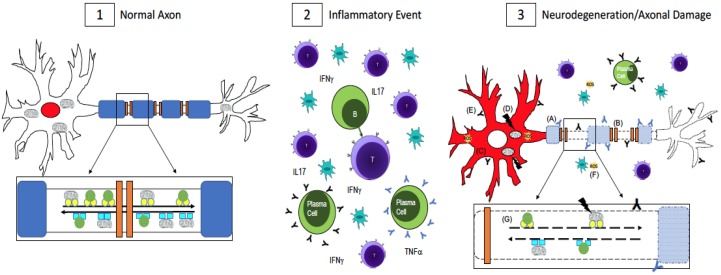
Axonal Damage in multiple sclerosis (MS). (**1**) In a normal, healthy axon, myelin (blue) wraps around the axon and ion channels (orange) are clustered in the unmyelinated nodes of Ranvier. This enables saltatory conduction for fast signal transmission down the axon. RNA binding proteins, which maintain RNA homeostasis, are localized to the nucleus (red). Mitochondria and other cargo (green circles) are transported retrogradely along the axon by dynein (turquoise squares, inset) while anterograde transport is done by kinesin motor proteins (yellow circles, inset). Transport is fast and uninterrupted because the axon is undamaged, has no energy shortage, and the motor proteins are intact; (**2**) In MS, there is central nervous system infiltration of T-cells, B-cells, plasma cells, and macrophages, which lead to a cascade of events including the release of pro-inflammatory cytokines and antibodies, which are thought to be harmful to both myelin and neurons and axons. [43,44,45,46,47,48,49]; (**3**) Axonal damage and neurodegeneration occur simultaneously with inflammation. There is ongoing demyelination due to antibodies against myelin antigens (blue Y, A) [43,44,45]. Demyelination leads to the redistribution of ion channels (B), which impairs conduction along the axon [32,33]. The redistribution of RNA binding proteins from their normal nuclear location (panel 1, red) to the cytoplasm (panel 3, C, red) is a pathological feature of neuronal degeneration in neurological diseases [25,26,27,28,30,31]. Mutations in mitochondrial DNA (D) can impair the cell’s ability to generate enough ATP while antibodies to non-myelin antigens (black Y, E) damage axons [46,47,48,49]. Reactive oxygen species (yellow, F) could be released from activated microglia or as a result of dysfunctional mitochondria [37,38,39,40,41]. Impaired fast axonal transport (G) is also evident in MS [9,34,35,36]. A combination of these events, as opposed to one in particular, contributes to neurodegeneration and axonal damage in MS.

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
