# Peer review of "Contribution of the Degeneration of the Neuro-Axonal Unit to the Pathogenesis of Multiple Sclerosis"

_brainsci, 2017, doi:10.3390/brainsci7060069_

Round 1
Reviewer 1 Report
In this review the authors tried to summarize some of the mechanisms probably involved in neurodegeneration and axon damage responsible for disability of multiple sclerosis affected patients. Even if the almost all the relevant references are included in this review in my opinion the authors should add also this paper:
Yukitake M, Sueoka E, Sueoka-Aragane N, Sato A, Ohashi H, Yakushiji Y, Saito M, Osame M, Izumo S, Kuroda Y. Significantly increased antibody response to heterogeneous nuclear ribonucleoproteins in cerebrospinal fluid of multiple sclerosis patients but not in patients with human T-lymphotropic virus type I-associated myelopathy/tropical spastic paraparesis. J Neurovirol. 2008 Apr;14(2):130-5. doi: 10.1080/13550280701883840. PubMed PMID: 18444084
Author Response
We thank the reviewer for their comments and have made the following changes:
· We have added the suggested citation to the manuscript (lines 216, 591-595).
Reviewer 2 Report
The title is a bit vague. Neurodegeneration in itself is vague. A more descriptive title would be beneficial. In the introduction more detail would be beneficial. 50% of RRMS progress to SPMS at 10 years, why this happens is not known. There is no mention of RIS or CIS. "A smaller portion", that number is generally accepted as 5%. SPMS patients RARELY experience "recovery" periods, that's why the disease is described as "progressive". There should be better citation for the antibody contribution in reference to Figure 1. The "neurodegeneration" could also result from CD8 or NK cell contributions. Also Neutrophils may play a part.
On page 5 lines 190 -200 and 227-237, these areas read too much like a report, lots of technical, methodological details. For a review, the language should be less technical..."We did..." etc.
Author Response
We thank the reviewer for their comments and have made the following changes:
· We recognize that the title and the term ‘neurodegeneration’ may be vague and as we are specifically highlighting neuronal and axonal damage in this review, we have changed the title to ‘Contribution of the degeneration of the neuro-axonal unit to the pathogenesis of multiple sclerosis’ (lines 2,3).
· We have also expanded the introduction to make it more complete (lines 26, 28, 30-33).
· We apologize for the lack of citations in the figure and have fixed this error (lines 113,115,116,118,120-122).
· We have added citations noting the contributions of CD8 and NK cells (lines 178-181, 518-526).
· With regards to the technical language on page 5 lines 190-200 and 227-237, we appreciate this reviewer’s concerns. However, we believe that this language accurately describes the bench work performed in the lab and is appropriate considering the complexity of the work, which is needed to describe the functions of hnRNP A1 and its role in neurodegeneration.
Round 2
Reviewer 2 Report
This is an improved manuscript.